# Effects of Physical Activity and Circadian Rhythm Differences on the Mental Health of College Students in Schools Closed by COVID-19

**DOI:** 10.3390/ijerph20010095

**Published:** 2022-12-21

**Authors:** Huimin Li, Yong Zhang

**Affiliations:** Physical Education Institute, Anhui Polytechnic University, Wuhu 241000, China

**Keywords:** college students, physical activity levels, circadian rhythm differences, psychological disorders, sequestration management

## Abstract

Purpose: Since the prolonged sequestration management that was implemented in order to achieve lower infection and mortality rates, there has been a surge in depression worldwide. The correlation between the physical activity level and the detection rate of a depressed mood in college students should be of wide concern. A large number of studies have focused on the association between physical activity levels and a negative mood, but circadian rhythm differences seem to be strongly associated with both physical activity levels and mental illness. Therefore, this paper will examine the correlation between physical activity levels, circadian rhythm differences, and mental health levels in college students. METHODS: Data were collected through a web-based cross-sectional survey. In June and December 2022, questionnaires were administered to college students from three universities in Anhui, China. In addition to socio-demographic information, measures included the International Physical Activity Questionnaire-Short Form (IPAQ-SF), Morning and Evening Questionnaire-5 Items (MEQ-5), and Symptom Check List90 (SCL-90) scales. Correlation analysis was used to understand the relationship between physical activity and circadian rhythm differences in the three aspects of college student’s mental health. RESULTS: The analysis of the data led to the conclusion that 28.4% of the 1241 college students in this survey had psychological disorders. The physical activity level of male students was higher than that of female students, but the risk of having depressive tendencies was higher in female students than in male students. There was a significant negative correlation between the physical activity level and scl-90 scores (*p* < 0.01), which indicates that higher physical activity levels are associated with higher mental health. Circadian rhythm differences and scl-90 scores were significantly positively correlated among college students (*p* < 0.01), and night-type people had a higher risk of mental illness than intermediate-type and early-morning-type people. CONCLUSIONS: During the period of closed administration due to COVID-19, school college students experienced large and high levels of negative emotional phenomena due to reduced physical activity and public health emergencies. This study showed significant correlations between both physical activity levels and circadian rhythmicity differences and the degree of mental health of college students.

## 1. Introduction

Since the outbreak of the Corona Virus Disease 2019, various countries have developed a series of prevention and control measures to control the growth of the epidemic and reduce infection and mortality rates. This initiative has been linked to an increase in the prevalence of major depressive disorder (MDD), which has been noted worldwide through reducing the movement of people to contain the source of infection. In particular, women were more affected by the epidemic than men, and younger age groups were affected more than older adults. The global public healthcare burden of mental disorders has become heavier in the wake of the new crown epidemic, with data showing a surge in the number of people with depression of 53 million, an increase of 27.6% [1]. In China, college students, as a population-based, dense, and highly mobile group, had to take closed-management measures in schools during the outbreak. During the epidemic closure control, students were asked not to leave school unless necessary, and some universities even adopted an online teaching model, which greatly reduced the physical activity level of college students in school. The reduction in physical activity was shown to be associated with the outbreak of depression during COVID-19. Physical activity is one of the most important sources of physical and mental health, and more physical activity is positively associated with greater well-being compared to a lack of physical activity and sedentary behaviors, which increase conditions such as anxiety and depression [2,3,4]. In contrast, college students, as a special group transitioning from adolescence to adulthood, are faced with sudden major public health events that strongly affect not only their physical health but also their mental health and collective behavior. Therefore, in recent years, more and more college students have been experiencing mental health problems. In the 2022 National Depression Blue Book report, it was revealed that fifty percent of the nation’s depressed patients are school students, and the prevalence of 18–24-year-olds accounts for 35.32% of the nation’s patients.

A growing number of researchers now believe that circadian rhythm disturbances may be the mechanism of illness in a proportion of depressed individuals [5,6]. The circadian system refers to the activity of organisms on a 24-h cycle, and this system influences and regulates the timing of almost all human behavior and physiology [7]. Studies in adolescence and early adulthood have found that night-time populations are more likely to suffer from depression than early-morning populations and that there is a higher risk of suicide among night-time populations [8]. In addition, motor activity is an important circadian input [9]. Rodent models show that physical activity increases the amplitude of circadian rhythm markers and alters the phase of clock genes in peripheral tissues. Physical exercise in humans has also been shown to alter circadian rhythm markers, including body temperature and melatonin rhythms [10,11]. It is hypothesized that humans with a biologically abnormal component in the circadian system may be more susceptible to depression [12,13].

Physical activity has been suggested to have antidepressant effects [14]. In the analysis of the effects of physical activity on mental health, there is a large body of research showing the positive effect of physical activity on changes in mood produced by environmental stimuli. Performing physical exercise can produce benign changes in psychological qualities, such as enhancing subjective well-being, improving individual self-confidence, and making participants happy and optimistic [15]. There is also a beneficial effect on the physiological level of the individual, and Madrigal et al. suggest that physical activity can improve the physical health and physical state of an individual, promote positive changes such as dopamine hormone secretion, and promote the overall mental health of an individual [16].

In the existing studies on the relationship between physical activity and psychological problems, there are more articles on the relationship between physical activity and psychological problems, and no studies on the link between physical activity, circadian rhythmicity differences, and psychology have been found. In this paper, we will discuss the impact of the differences between physical activity levels and circadian rhythmicity on the mental health of college students.

## 2. Materials and Methods

### 2.1. Participants in the Survey

This study was conducted using an online completed questionnaire to investigate university students who were enrolled in schools and cities with closed management measures due to the COVID-19 outbreak. The questionnaires were distributed to three universities in Anhui Province, China, in the freshman, sophomore, junior, and senior years, with a total of 1241 participants, 602 male and 639 female. The questionnaires were administered in the classroom under the supervision and guidance of teachers to ensure the validity of the responses. The questionnaire consisted of three scales: The International Physical Activity Questionnaire-Short Form (IPAQ-SF), Morning and Evening Questionnaire-5 Items (MEQ-5), and Symptom Check List90 (SCL-90). Socio-demographic information included age, gender, grade, height, and weight.

#### 2.1.1. The International Physical Activity Questionnaire-Short Form (IPAQ-SF)

The International Physical Activity Questionnaire (2002) is a 7-day-long self-report measure of physical activity recall, consisting of both a long questionnaire and a short questionnaire, suitable for use with adolescents and adults (15–69 years). The present study used the ipaq-sf, which has been tested in 12 countries, both developed and developing, and has shown acceptable reliability and validity properties in both [17]. Briefly, the ipaq-sf asks participants to report the frequency and intensity of physical activity over the past 7 days, as well as the amount of time typically spent on these physical activities each day, to the nearest hour and minute. The questionnaire classifies physical activity levels as high intensity (MET-min/w > 3000), moderate intensity (600 < MET-min/w < 3000), and low intensity (MET-min/<600). 

#### 2.1.2. Morning and Evening Questionnaire-5 Items (MEQ-5)

The Morning and Evening Questionnaire, developed by Home and Ostberg in 1976, is a 19-item (MEQ-19), and a more concise 5-item (MEQ-5), scale that has been widely used around the world to assess circadian rhythms. The validity and reliability of the Chinese version of the MEQ-5 used in this study have been validated [18]. The circadian rhythms of the testers were classified as absolute night type (4–7), moderate night type (8–11), intermediate type (12–17), moderate early morning type (18–21), and absolute early morning type (22–25) by summing up the scores of five questions.

#### 2.1.3. Symptom Check List 90 (SCL-90)

In 1975, L.R. Derogatis developed Symptom Check List 90. The Scl-90 is widely used to measure clinical psychiatric symptoms and mental health status [19,20] and is intended for people 16 years of age and older. This scale requires the test taker to recall and assess their psychological and physical self over a week and to respond to 90 questions from a proprioceptive perspective. Each item is rated on a scale of 1 to 5, as follows: 1 not present: self-consciousness of the problem (symptom); 2 very mild: self-consciousness of the problem, but not frequent and severe; 3 moderate: self-consciousness of the symptom, and its severity is mild to moderate; 4 severe: self-consciousness of the symptom, and its severity is moderate to severe; 5 serious: self-consciousness of the symptom is very severe in terms of frequency and intensity. The Scl-90 scale can be used to determine the presence or severity of psychological disorders by three criteria. In this study, people with a total score of more than 160 were selected as tending to have psychological disorders.

## 3. Results

### 3.1. Gender

The analysis of gender and scl90 scores in 1241 questionnaires showed a positive correlation with *p* < 0.01 (Table 1), which was significant. This questionnaire screened a total of 353 students with psychological disorders, accounting for 28.4% of the total number of students. Among them, 141 were male students, accounting for 39.94% of the population with psychological disorders, while the number of female students was 212, accounting for 60.05%.

Analysis of the total number of physical activity levels and gender revealed a statistically significant negative correlation at *p* < 0.01 (Table 1). Further analysis of the physical activity level and gender of the 353 individuals with psychological disorders revealed that the gender and physical activity index were significantly positively correlated, and *p* < 0.01 was statistically significant (Table 2).

### 3.2. Physical Activity Level

The correlation between physical activity level scores and scl-90 scores of the total number of students showed a significant negative correlation with *p* < 0.01 (Table 1); this finding was also reflected in the 353 students with mental disorders, as shown in Table 2, with a significant negative correlation with *p* < 0.01 by Spearman’s analysis. This indicates that when the level of physical activity of college students is higher, the risk of suffering from psychological disorders is lower.

### 3.3. Circadian Rhythmicity Differences

In the total sample size, the correlation analysis of early-morning-night-type scores and scl90 scores yielded *p* < 0.01, a statistically significant negative correlation (Table 1). The same conclusion was also reached in the 353 individuals with scl-90 scores greater than 160 (*p* = 0.002, *p* < 0.01). This indicates that when the score of the circadian rhythm is lower, the score of scl-90 is higher. Due to the premise that the early-morning-night-type is classified by the lowest score for the night type and the highest for the early morning type, this seems to imply that the night type population is more likely to have psychological problems than the intermediate and early morning types.

Although night types have been shown in other studies to have lower physical activity levels than early morning types, this was not reflected in the present study. An analysis comparing physical activity and circadian rhythm types yielded a negative correlation, but one with *p* > 0.05, which was not statistically significant.

## 4. Discussion

**Main finding**: This article examines the effects of physical activity and circadian rhythmicity differences on the mental health of college students in closed schools. It was found that female college students had lower levels of physical activity and a higher risk of developing mental illness than male college students, while male college students had higher levels of physical activity and a lower tendency to develop depression compared to female students during the closure of epidemic-controlled schools. A comparative analysis of circadian rhythmicity and scl-90 scores also led to the conclusion that late sleepers were more likely to have mental health problems among college students in schools that were closed due to the epidemic. The above findings confirm that physical exercise not only has a beneficial effect on the physical health of university students during epidemic closure but also alleviates their anxiety and depression to a certain extent. Continuing to pay attention to the mental health status of university students during epidemic closure management, a timely prevention, and improving psychological conditions through changes in living and sleeping habits and exercise interventions would be the least costly and most efficient means of intervention.

**Compared to past research:** past studies have shown that exercise and physical activity can have beneficial effects on depression, even comparable to the effects of medication [21,22]. Exercise interventions in hospitalized adolescent depressed patients were found to reduce depressive symptoms, but no beneficial effects were found in anxiety [23]. Physical activity is effective in reducing depression and anxiety in college student populations [24,25], and even at levels below public health recommendations, physical activity has significant benefits for mental health [26]. The relationship between physical activity, circadian rhythm differences, and mental health has not been investigated in previous studies, although the effects on mental health of physical activity levels and circadian rhythm differences on their own are widely available. The starting point of this study would be to find better ways to treat mental illness by changing the markers of circadian rhythms through exercise.

**Limitations:** The questionnaire for this study was not collected for college students nationwide at the time of collection. Data will continue to be collected at a later date to follow up on the effects of differences in physical activity levels and circadian discipline on mental health for the same sample size and to analyze the comparative results. Future research directions could focus on the extent to which physical activity of different intensities alters human circadian rhythm markers and a related study of the effects of such alterations on mental health.

**Practical implications:** According to the World Health Organization, approximately one billion people worldwide are suffering from mental disorders, and one person loses his or her life to suicide every 40 s. In recent years, student suicides due to depression have occurred repeatedly, and this includes elementary school students at an early age. The gradual youthfulness of the depression suicide rate is not only heartbreaking but also raises heavy thoughts on how to prevent and treat it. According to relevant data in China, 52% of patients do not consider psychological treatment because they cannot afford the expensive costs. Additionally, under the control of the epidemic, 43% of depression patients have to change their way of seeking medical treatment. In this environment, physical activity has become convenient and easy to maintain, with little to no monetary cost, and it can have a beneficial impact on both physical and mental health. This study shows that belonging to the night type is more likely to trigger the emergence of psychological problems. Two interventions, sleep management and physical exercise, can prevent and improve the negative mood of college students, which seems to be the best complementary treatment for college students with psychological disorders who cannot pay for medical care and cannot be seen in time due to epidemic control.

## 5. Conclusions

During the period of closed administration due to COVID-19, school college students experienced large and high levels of negative emotional phenomena caused by reduced physical activity and unexpected public health events. Physical activity and circadian rhythm differences were two of the factors that created psychological disturbances among college students during this period. This study showed significant correlations between both physical activity levels and circadian rhythmicity differences and the degree of mental health of college students. High levels of physical activity can effectively alleviate negative emotions caused in college students during closed administrations due to COVID-19. A continued attention to the mental health status of university students under closure management during the epidemic and the timely prevention and improvement of psychological conditions through changes in lifestyle, sleep habits and exercise are cost-effective interventions.

## Figures and Tables

**Table 1 ijerph-20-00095-t001:** Correlation between gender, physical activity, circadian rhythm differences, and SCL-90 in the total population N = 1241.

		Physical Activity	Scl-90	Gender	Morning Night Type
Physical activity	Spearman Sig. (2-tailed)	1.000	−0.065 * 0.022	−0.133 ** 0.000	−0.015 0.606
Scl-90	Spearman Sig. (2-tailed)	−0.065 * 0.022	1.000	0.135 ** 0.000	−0.117 ** 0.000
Gender	Spearman Sig. (2-tailed)	−0.133 ** 0.000	0.135 ** 0.000	1.000	−0.007 0.800
Morning night type	Spearman Sig. (2-tailed)	−0.015 0.606	−0.117 ** 0.000	−0.007 0.800	1.000

Notice: ** At the 0.01 level (two-tailed), the correlation is significant. * At the 0.05 level (two-tailed), the correlation is significant.

**Table 2 ijerph-20-00095-t002:** Correlation between gender, physical activity, circadian rhythm differences, and scl-90in people with psychological disorders N = 353.

		Physical Activity	Scl-90	Gender	Morning Night Type
Physical activity	Spearman Sig. (2-tailed)	1.000	−1.45 ** 0.006	−0.116 * 0.029	−0.080 0.136
Scl-90	Spearman Sig. (2-tailed)	−0.145 ** 0.006	1.000	−0.043 0.422	−0.168 ** 0.02
Gender	Spearman Sig. (2-tailed)	−0.116 * 0.029	−0.043 0.422	1.000	0.033 0.533
Morning night type	Spearman Sig. (2-tailed)	−0.080 0.136	−0.168 ** 0.02	0.033 0.533	1.000

Notice: ** At the 0.01 level (two-tailed), the correlation is significant. * At the 0.05 level (two-tailed), the correlation is significant.

## Data Availability

Link: https://pan.baidu.com/s/1bMGTDxCvYJtBiIgeaDNL8Q (accessed on 10 December 2022). Extraction code: cc41.

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
