# Peer review of "Effects of Physical Activity and Circadian Rhythm Differences on the Mental Health of College Students in Schools Closed by COVID-19"

_ijerph, 2022, doi:10.3390/ijerph20010095_

Round 1
Reviewer 1 Report
Dear Authors,
the Manuscript Effects of physical activity and circadian rhythm differences on the mental health of college students closed by the COVID-19 is a questionnaire composed of IPAQ-SF, MEQ-5 and SCL-90 and including 601 students. The main drawback is the self-reporting of the physical activity. In addition, frequency of participants per groups according to circadian rhythm were low in case of early morning and late night types, while the majority was in the intermediate category, there caution should be applied when analyzing the data. Moreover, it is difficult to conclude the causal relationship between the level of physical activity and the depression index- could it be vice versa?
It would be nice to have more relevant sample sizes.
Author Response
Dear Reviewer:
Thank you for your detailed revision of my manuscript in your busy schedule. I will respond to the issues you pointed out in the following points:
- The MEQ-5 questionnaire used in the body report has been confirmed in several papers to be used to assess individual physical activity with some reliability and validity, and I have marked it in this revision of the manuscript. In addition, I checked the relevant literature of this journal and found that most of the questionnaires used by the authors to measure physical activity are MEQ-5.
- The point you made that it would be good to increase the number of questionnaires, we continued to distribute questionnaires to university students during these days, and 640 valid questionnaires were returned, adding to the previous total of 1241 questionnaires, and the conclusions drawn from the analysis and comparison have changed for the better.
- The correlation between circadian rhythmicity and scl-90 scores was analyzed after increasing the sample size and showed a significant negative correlation, p < 0.01.
- The correlation between physical activity level and depression was also confirmed several times. Similarly, in the current study, physical activity level and scl-90 scores were also significantly negatively correlated (p < 0.01), which means that when physical activity level is lower, scl-90 scores are higher.
Thank you for your patience and have a nice life!
Kind regards,
Huiminli
Reviewer 2 Report
The material presented for review touches on very important and current aspects of health related to the pandemic and the isolation of people. Low physical activity is a serious problem in highly developed countries. It affects practically everyone - children, adolescents, adults and the elderly.
The authors, wh
Please check the compliance of the prepared manuscript with the editorial requirements of the journal
en addressing the impact of physical activity and differences in circadian rhythms on the mental health of students closed by COVID-19, show empathy and social sensitivity. The reviewer's attention concerns the fact that when studying a group (n =601) of students but only one university, in the selected city the results should not be generalized to all students - as the title of the manuscript suggests.
Author Response
Dear Reviewer:
Thank you for your detailed revision of my manuscript in your busy schedule. I will respond to the issues you pointed out in the following points:
- According to your suggestion I checked if the manuscript meets the requirements of the journal, and the manuscript was edited and uploaded using the word version required by the journal.
2. For the problem of distributing questionnaires to only one university, we sent questionnaires to two other universities in the past few days and got 640 valid questionnaires, and now the sample size is 1241 in total, and the analysis and comparison concluded that there is also a good change.Thank you for your patience and have a nice life!
Kind regards,
Huiminli